# PROVABLY EFFICIENT CONTEXTUAL LINEAR QUADRATIC REGULATOR

## ABSTRACT

A fundamental challenge in artificially intelligence is to build an agent that generalizes and adapts to *unseen* environments. A common strategy is to build a decoder that takes a context of the unseen new environment and generates a policy. The current paper studies how to build a decoder for the fundamental continuous control environment, linear quadratic regulator (LQR), which can model a wide range of real world physical environments. We present a simple algorithm for this problem, which uses upper confidence bound (UCB) to refine the estimate of the decoder and balance the exploration-exploitation trade-off. Theoretically, our algorithm enjoys a $\widetilde{O}\left(\sqrt{T}\right)$ regret bound in the online setting where $T$ is the number of environments the agent played. This also implies after playing $\widetilde{O}\left(1/\epsilon^2\right)$ environments, the agent is able to transfer the learned knowledge to obtain an $\epsilon$-suboptimal policy for an unseen environment. To our knowledge, this is first provably efficient algorithm to build a decoder in the continuous control setting. While our main focus is theoretical, we also present experiments that demonstrate the effectiveness of our algorithm.

## 1 INTRODUCTION

Humans are able to solve a new task *without any training* based on previous experience in similar tasks. Our intelligent agent should be able do the same, learning from previous experience, adapting to the new ones and improving the performance as the agent gains more experience. This is a challenging problem as we need to design an adaptation mechanism which is fundamentally different from classical supervised learning methods.

A common approach is to build a decoder so that once the agent sees a description of new task, i.e., the context of the new task, the decoder turns the context into a succinct representation of the new task, based on which the agent is able to design a policy to solve the task. Note this procedure resembles how a human solves a new task. For example, if a human wants to push an object on a table, the human first sees the object and the table (context). Then, in his/her mind, the context becomes a representation of this task, e.g., a sense of weight of the object. Based on this representation, the human can easily reason about how much force to exert on the object.

This general approach has been applied in practice. For example, Wu et al. (2018) studied the visual navigation task and built a Bayesian model that takes the context of new environments and outputs the policy that enables the agent to navigate. Killian et al. (2016) used this approach to develop personalized medicine policies for HIV treatment.

While this is a promising approach, currently we only have limited theoretical understanding. The approach can be formulated in Contextual Markov Decision Process (CMDP) framework (Hallak et al., 2015). Recently, there is a line of work gave provable guarantees for CMDP (Abbasi-Yadkori & Neu, 2014; Hallak et al., 2015; Dann et al., 2018; Modi et al., 2018; Modi & Tewari, 2019). These work all studied tabular MDPs, and use function approximation, e.g., linear functions, generalized linear models, etc, to model the mapping from the context to the probability transition matrix. A major drawback of these work is that they are restricted to the tabular setting and thus can only deal with discrete environments. Therefore, they can hardly model real-world continuous control tasks, like the task of pushsing an object as we described above. A natural question arises:

### Can we design a provably efficient decoder for continuous control problems?

In this paper, we make an important step towards answering this question. We study the fundamental setting in continuous control, linear quadratic regulator (LQR). LQR is arguably the most widely used framework in continuous control, as LQR easily models real world physical phenomena, e.g., the pushing object task we described earlier. We propose a new algorithm that builds a decoder, so that for a new LQR task, the decoder takes LQR's context and outputs a representation based on which the agent can easily infer a near-optimal policy for new continuous control tasks. In the training phase, we build the decoder via a sequence of LQRs (in an online fashion) with unknown parameters. For each new task, we first use the current decoder to build the representation of this task, infer a policy based on this representation and use this policy to do control for this episode. There are two crucial components in our algorithm. First, after each episode, we will refine the estimate of the decoder based on the observations from this episode. Second, it is crucial to use a upper confidence bound (UCB) estimator of the decoder to build the representation so that the agent can perform a near-optimal trade-off between exploration and exploitation. In this way, we provably show the decoder improves the performance as it experiences more training tasks. Formally, we show our algorithm enjoys $\widetilde{O}\left(\sqrt{T}\right)$ regret (the difference between the cumulative rewards of our algorithm and the unknown optimal policy on every seen environment) bound in the online setting. Moreover, the algorithm is able to obtain an $\epsilon$-suboptimal policy for an unseen LQR environment after playing $\widetilde{O}\left(\epsilon^{-2}\right)$ environments. To our knowledge, this is the first provably efficient algorithm that builds a decoder for continuous control environments. Empirically, we simulate several physical environments to illustrate the effectiveness of our algorithm.

**Organization**  This paper is organized as follows. In Section 2, we discuss related work. In Section 3, we formally describe the problem setup. In Section 4, we present our algorithm and its theoretical guarantees. In Section 5, we use simulation on physical environments to demonstrate the effectiveness of our approach. We conclude in Section 6 and defer most technical proofs to the appendix.

## 2 RELATED WORK

Recently there is a large body of literature focusing on learning for control in LQR systems. The first work we are aware of is Fiechter (1997) which studies the sample complexity of LQR in the offline setting. For the online setting, where the agent can only obtain the next state starting from the present state, the first near-optimal regret bound ($\widetilde{O}(\sqrt{T})$) is due to Abbasi-Yadkori & Szepesvári (2011), which studies the learning problem in the infinite-horizon average-case cost setting. Later on, a sequence of papers (Tu & Recht, 2017; Dean et al., 2017; 2018; Tu & Recht, 2018; Abbasi-Yadkori et al., 2018; Cohen et al., 2019) studied this problem in similar settings, improved efficiency of the algorithms and characterized the gap between model-free and model-based approaches.

Building an agent that quickly adapts to new environment has received increasing interest in the machine learning community. Taylor & Stone (2009) gave a summary for the literature status before 2009. More recently, a sequence of theory papers Lehnert & Littman (2018); Spector & Belongie (2018); Abel et al. (2018); Lehnert et al. (2019) studied the transferability of reward knowledge, state-abstraction, and model features for Markov decision processes. Please also refer to references in paper cited above for more details. There are also some experimental works, e.g., Santara et al. (2019); Yu et al. (2018); Wu et al. (2018); Gamrian & Goldberg (2018), studying how to transfer knowledge from seen tasks to unseen tasks. Nevertheless, we are not aware of any study on how to provably perform continuous control with contexts.

## 3 PRELIMINARIES

**Notations.**  We begin by introducing necessary notations. We write $[h]$ to denote the set $\{1, \ldots, h\}$. We use $I_d \in \mathbb{R}^{d \times d}$ to denote the $d$-dimensional identity matrix. We use $0_{d \times d'}$ to represent the all-zero matrix in $\mathbb{R}^{d \times d'}$. If it is clear from the context, we omit the subscript $d \times d'$. Let $\|\cdot\|_2$ denote the Euclidean norm of a vector in $\mathbb{R}^d$. For a symmetric matrix $A$, let $\|A\|_{\mathrm{op}}$ denote its operator

norm and $\lambda_i(A)$ denote its $i$-th eigenvalue. Throughout the paper, all sets are multisets, i.e., a single element can appear multiple times.

**Finite Horizon Linear Quadratic Regulator.** We now formally define the finite horizon Linear Quadratic Regulator (LQR) problem. In the LQR problem, there is a state space $\mathcal{X} \subset \mathbb{R}^d$ and a closed action space $\mathcal{U} \subset \mathbb{R}^{d'}$. Suppose we always start from the initial state $x_1 = x_{\text{init}} \in \mathcal{X}$ and play for $H$ steps. Then at a state $x_h \in \mathcal{X}$, if an action $u_h \in \mathcal{U}$ is played, the next state is given by

$$x_{h+1} = Ax_h + Bu_h + w_{h+1}, \tag{1}$$

where $A, B$ are matrices of proper dimension and $w_{h+1}$ is a zero-mean random vector. Here $A, B$ can be viewed as the succinct representation of this LQR because as will be explained below, given $A, B$, we can easily infer the optimal policy for this LQR. For simplicity, we additionally denote

$$M = [A, B], \text{ and } y_h = [x_h^\top, u_h^\top]^\top \in \mathbb{R}^{d+d'}.$$

Now the state transition can be rewritten as $x_{h+1} = My_h + w_{h+1}$. For the ease of presentation, we assume that the covariance matrix of noise vector $w_{h+1}$ is $\mathbb{E}(w_{h+1}w_{h+1}^\top) = I_d$. Our analysis follows similarly if the covariance matrix is not $I_d$ (see e.g. Remark 3 of Abbasi-Yadkori & Szepesvári (2011)). After each step, the player receives an immediate cost $x_h^\top Q_h x_h + u_h^\top R_h u_h$, where $Q_h, R_h$ are positive definite (PD) matrices of proper dimensions. Throughout the paper, we assume the agent knows $Q_h$ and $R_h$ for all $h$. At a terminal state $x_H$, there is no action to be played, and the player receives a terminal cost $x_H^\top Q_H x_H$, where $Q_H$ is a PD matrix of proper dimension. The goal of the player is to find a *policy* $\pi : (\mathcal{X} \times \mathcal{U})^* \times \mathcal{X} \to \mathcal{U}$, which is a function that maps the trajectory $\{(x_i, u_i)\}_{i=1}^{h-1} \cup \{x_h\}$ to the next action $u_h$, such that the following objectives are minimized:

$$\left\{ J_h^\pi(M, x) := \mathbb{E}\left[ \left( \sum_{h'=h}^{H-1} x_h^\top Q_h x_h + u_h^\top R_h u_h \right) + x_H^\top Q_f x_H \ \middle| \ x_h = x \right] \right\}_{h \in [H]},$$

where the action $u_h$ is given by $u_h = \pi[(x_1, u_1), (x_2, u_2), \ldots, (x_{h-1}, u_{h-1}), x_h]$, and the expectation is over the randomness of $w_h$ and $\pi$.

It is well-known that the optimal policy $\pi^*$ is Markovian Puterman (2014), i.e., it only depends on the present state. For an unconstrained action space $\mathcal{U}$, we have

$$\forall x \in \mathcal{X}, h \in [H-1]: \quad \pi_h^*(M, x) := K_h(M)x$$

where $M = [A, B]$ and $K_h(M)$ is a matrix that will be defined shortly. It is also known (see e.g. Bertsekas (1996)) that the optimal cost function $J_h^*(x) := J_h^{\pi^*}(x)$ is given by

$$J_h^*(M, x) := x^\top P_h(M)x + C_h(M) = \inf_\pi J_h^\pi(M, x) \tag{2}$$

where

$$P_h(M) = \begin{cases} Q_h + A^\top P_{h+1}(M)A - A^\top P_{h+1}B(R_h + B^\top P_{h+1}(M)B)^{-1}B^\top P_{h+1}(M)A & h < H \\ Q_H & h = H \end{cases} \tag{3}$$

and

$$C_h(M) = \begin{cases} C_{h+1}(M) + \mathbb{E}_{w_{h+1}}\left[w_{h+1}^\top P_{h+1}(M)w_{h+1}\right] & h < H \\ 0 & h = H \end{cases}.$$

We now define $K_h(M)$ as

$$K_h(M) := -(R_h + B^\top P_{h+1}(M)B)^{-1}B^\top P_{h+1}(M)A. \tag{4}$$

Note that the optimal value Equation (2) satisfies Bellman equations,

$$\forall h \in [H-1]: \quad J_h^*(M, x) = x^\top Q_h x + \pi^*(x)^\top R_h \pi^*(x) + \mathbb{E}\left[J_{h+1}^*(Ax + B\pi^*(x) + w)\right]$$

and

$$\forall h \in [H-1]: \quad J_h^*(M, x) = x^\top Q_h x + \min_u \mathbb{E}[u^\top R_h u + J_{h+1}^*(Ax + Bu + w)].$$

Now we have shown that if we are given $A$ and $B$, then we can obtain the optimal policy directly. In this paper, we deal with setting where $A$ and $B$ are *unknown* and we need to use decoder to decode $A$ and $B$ from the contexts of the current LQR, as specified below.

**Learning to Control LQR with Contexts**   In the continuous control with contexts setting, in each episode we observe a context

$$(C, D) \sim \mu,$$

where $\mu$ is a distribution on $\mathbb{R}^{p \times d} \times \mathbb{R}^{p' \times d'}$. The context $[C, D]$ encodes the information of the environment. Formally, the representation $([A, B])$ of this environment can be decoded from the context via a decoding matrix $\Theta_* \in \mathbb{R}^{d \times (p+p')}$:

$$[A, B] = \Theta_* \cdot \begin{bmatrix} C & 0_{p \times d'} \\ 0_{p' \times d} & D \end{bmatrix}. \tag{5}$$

From now on, to emphasize that the representation of LQR can be decoded from $\Theta_*$, we write

$$M_{\Theta_*, C, D} := \Theta_* \cdot \begin{bmatrix} C & 0_{p \times d'} \\ 0_{p' \times d} & D \end{bmatrix} = [A, B]. \tag{6}$$

If it is clear from the context, we ignore $[C, D]$ for notational simplicity. Note the optimal decoder $\Theta_*$ is unknown to the agent and the goal is to learn $\Theta_*$ from contexts and interactions with the environment. Below we formally define the problem that we study.

**Definition 3.1** (Contextual Transfer Learning Problem). *Build an agent that plays on $K$ LQR games (one trajectory per game) with context pairs $\{(C^{(1)}, D^{(1)}), (C^{(2)}, D^{(2)}), \dots, (C^{(K)}, D^{(K)})\} \sim \mu$, for some integer $K \geq 0$ such that for another new context pair $(C, D) \sim \mu$, the agent outputs a policy $\pi$ based on $(C, D)$ which satisfies*

$$\mathbb{E}[J_h^\pi(M_{\Theta_*, C, D}, \ x_1) - J_h^*(M_{\Theta_*, C, D}, \ x_1)] \leq \epsilon$$

*for some given target accuracy $\epsilon > 0$.*

Here $K$ is the sample complexity which ideally scales *polynomially* with $\epsilon$ and problem-dependent parameters. The performance of the agent can also be measured by regret, as defined below.

$$\text{Regret}(KH) := \sum_{k=1}^{K} J_1^{\tilde{\pi}^{(k)}} \left( M_{\Theta_*, C^{(k)}, D^{(k)}}, \ x_1 \right) - J_1^* \left( M_{\Theta_*, C^{(k)}, D^{(k)}}, x_1 \right), \tag{7}$$

where $\tilde{\pi}^{(k)}$ is the policy played at episode $k$ by the agent. This quantity measurse the sub-optimality of policies the agent played in the first $K$ episodes.

**Remark 3.1.** *We consider matrix-type linear maps from context to the representation only for sake of presentation. Our algorithm and analysis can be readily extended to other linear maps, e.g., $[A_*(C), B_*(D)] := f(C, D)$ for some unknown linear function $f$.*

# 4   MAIN ALGORITHM

In this section, we first describe the algorithm and then present its sample complexity guarantees.

Since the decoder is linear, a straightforward algorithm is first to estimate $(A, B)$ using the trajectory from the single episode with system identification techniques (Mehra, 1974), and then to use linear regression to estimate the mapping from $(C, D)$ to $(A, B)$. However, this naive algorithm has two drawbacks. First, estimating $(A, B)$ accurately requires a long horizon. However, in our setup, we do not have any restrictions on the horizon. To fix this, we stack contexts and observations to construct a more direct estimate on the decoder (cf. Equation equation 8). Second, in order to achieve $\sqrt{T}$-type regret guarantee, one needs to balance exploration and exploitation carefully, but the naive algorithm does not have such an component. Our algorithm uses UCB to construct a confidence set which helps balance exploration and exploitation.

**Algorithm**   We describe the high-level idea of the algorithm below. The agent maintains a decoder that maps the context $(C, D)$ to the representation $(A, B)$. We denote $\Theta^{(k)}$ the decoder at the $k$-th episode. Initially, we know nothing about $\Theta_*$, so we initialize our decoder by setting $\Theta^{(1)} = 0 \in \mathbb{R}^{d \times p}$. At the $k$-th episode, the agent plays policy $\pi^{(k)}$ and in each time step $h \in [H-1]$, it collects data

$$x_h^{(k)}, u_h^{(k)}, x_{h+1}^{(k)}, \quad z_h^{(k)} \leftarrow \begin{bmatrix} C^{(k)} x_h^{(k)} \\ D^{(k)} u_h^{(k)} \end{bmatrix},$$

---

**Algorithm 1** Linear Continuous Control with Contexts

---

1: **Input** Total number of episodes $K$;
2: **Initialize** $\Theta^{(1)} \leftarrow 0 \in \mathbb{R}^{d \times 2p}$, $V^{(1)} \leftarrow I_{2p,2p}$, $W^{(1)} \leftarrow 0 \in \mathbb{R}^{2p \times d}$;
3: **for** episode $k = 1, 2, \ldots, K$ **do**
4:     Let $x_1^{(k)} \leftarrow x_{\text{init}}$, $V^{(k+1)} \leftarrow V^{(k)}$, $W^{(k+1)} \leftarrow W^{(k)}$;
5:     Obtain context $[C^{(k)}, D^{(k)}] \sim \mu$;
6:     Solve for the present policy:

$$\widetilde{\Theta}^{(k)} = \arg \min_{\Theta \in \mathcal{C}^{(k)}} J_1^* \left( M_{\Theta, C^{(k)}, D^{(k)}}, x_1^{(k)} \right) \qquad (10)$$

7:     where $J_1^*$ is given by Equation 2, and $\mathcal{C}^{(k)}$ is defined in Equation 11;
8:     **for** stage $h = 1, 2, \ldots, H - 1$ **do**
9:         Let the current state be $x_h^{(k)}$;
10:        Play action $u_h^{(k)} \leftarrow K_h \left( M_{\Theta^{(k)}, C^{(k)}, D^{(k)}} \right) \cdot x_h^{(k)}$, where $K_h$ is defined in Equation 4;
11:        Obtain the next state $x_{h+1}^{(k)}$;
12:        Let $z_h^{(k)} \leftarrow \begin{bmatrix} C^{(k)} x_h^{(k)} \\ D^{(k)} u_h^{(k)} \end{bmatrix}$;
13:        Update: $V^{(k+1)} \leftarrow V^{(k+1)} + z_h^{(k)} z_h^{(k)\top}$;
14:        Update: $W^{(k+1)} \leftarrow W^{(k+1)} + z_h^{(k)} \left( x_{h+1}^{(k)} \right)^\top$;
15:     Compute $\Theta^{(k+1)\top} \leftarrow \left( V^{(k+1)} \right)^{-1} W^{(k+1)}$;
16: **output** $\widetilde{\Theta}^{(k)}$ where $k$ is chosen from $[K]$ uniformly at random.

---

where $z_h^{(k)}$ can be viewed as the *context regularized* observation. We now describe how to obtain policy $\pi^{(k)}$. We first solve the following optimization problem,

$$\widetilde{\Theta}^{(k)} = \arg \min_{\Theta \in \mathcal{C}^{(k)}} J_1^* \left( M_{\Theta, C^{(k)}, D^{(k)}}, x_1^{(k)} \right)$$

where $J_1^*$ is given by Equation (2), and the confidence set $\mathcal{C}^{(k)}$ will be defined shortly. $\mathcal{C}^{(k)}$ represents our confidence region on $\Theta_*$. Since we choose the one that minimizes the cost, this represents the principle "optimism in the face of uncertainty" and it is the key to balance exploration and exploitation which will be more clear in the proof. Notice that the above optimization problem is a polynomial optimization problem. Then the policy is given by

$$\pi_h^{(k)}(x) := K_h \left( M^{(k)} \right) \cdot x \text{ where } M^{(k)} = M_{\Theta^{(k)}, C^{(k)}, D^{(k)}} := \Theta^{(k)} \cdot \begin{bmatrix} C^{(k)} & 0 \\ 0 & D^{(k)} \end{bmatrix},$$

and $K_h$ is given by Equation (4). After episode $k \in [K]$, we use the following ridge regression formulation to update decoder

$$\Theta^{(k)} = \left( \left( V^{(k+1)} \right)^{-1} W^{(k+1)} \right)^\top. \qquad (8)$$

where

$$V^{(k+1)} = I + \sum_{k'=1}^{k} \sum_{h=1}^{H-1} z_h^{(k')} z_h^{(k')\top} \quad \text{and} \quad W^{(k+1)} = \sum_{k'=1}^{k} \sum_{h=1}^{H-1} z_h^{(k')} x_{h+1}^{(k')\top}.$$

After playing $K$ episodes, the algorithm outputs a $\widetilde{\Theta}$ by picking one from $\{\widetilde{\Theta}^{(k)}\}_{k \in [K]}$ uniformly at random. Now for a new task with its context, our learned policy map is given by:

$$\forall C, D \sim \mu, x \in \mathcal{X}, h \in [H-1]: \quad \widetilde{\pi}_{C,D,h}(x) = K_h \left( \widetilde{\Theta} \cdot \begin{bmatrix} C & 0 \\ 0 & D \end{bmatrix} \right) \cdot x. \qquad (9)$$

The formal algorithm is presented in Algorithm 1.

## 4.1 Algorithm Analysis

To present the analysis of the algorithm, we first introduce some assumptions.

**Assumption 4.1.** *The contexts and LQR satisfy the following properties.*

- $\forall h \in [H], \|P_h(M)\|_2 \leq c_q$ *for some parameter* $c_q > 0$.
- $\|\Theta_*\|_F \leq c_\Theta$;
- $\forall h \in [2, H], i \in [d] : \quad \|w_h\|_2 < \infty$ *and* $\forall \gamma > 0, \mathbb{E}[\gamma w_{h,i}] \leq \exp(\gamma^2 c_w^2/2)$;
- $\forall x \in \mathcal{X}, u \in \mathcal{U}, (C, D) \in \text{supp}(\mu) : \quad \|Cx\|_2 + \|Du\|_2^2 \leq c_x^2, \|x\|^2 + \|u\|_2^2 \leq c_x^2$;
- $\forall (C, D) \in \text{supp}(\mu), x \in \mathcal{X}, h \in [H]: K_h(M_{\Theta_*,C,D}) \cdot x \in \mathcal{U}$.

*where* $c_\Theta, c_w, c_x$ *are some positive parameters.*

The first assumption is standard to ensure controllability. The second is a regularity condition on the optimal decoder $\Theta_*$. The third assumption imposes almost sure boundedness of the noise $w$. The fourth assumption is a regularity condition on the observation. The last assumption guarantees the optimal controller for the unconstrained LQR problem is realizable in our control set $\mathcal{U}$. Given these assumptions, We are now ready to define confidence set $\mathcal{C}^{(k)}$ as follows.

$$\mathcal{C}^{(k)} = \Big\{ \Theta : \ \text{tr}\big[(\Theta - \Theta^{(k)})V^{(k)}(\Theta - \Theta^{(k)})^\top\big] \leq \beta^{(k)},$$

$$\text{and } \forall h \in [H], (C, D) \in \text{supp}(\mu), \ \big\|P_h\big(M_{\Theta,C,D}\big)\big\|_2 \leq c_q \Big\}, \tag{11}$$

where $P_h$ is given by Equation (3) and $\beta^{(k)}$ is defined as follows,

$$\beta^{(k)} = \Big( c_\Theta + c_w \sqrt{2d\big( \log d + p \log(1 + kHc_x^2/p)/2 + \log \delta^{-1}\big)} \Big)^2. \tag{12}$$

We remark that $\mathcal{C}^{(k)}$ is changing at every episode because we update $\Theta^{(k)}$ and $V^{(k)}$ at every episode. The size of $\mathcal{C}^{(k)}$ is decreasing because $V^{(k)}$ is strictly increasing at every episode.

With the above assumptions, the guarantee of Algorithm 1 is formally presented in the next theorem.

**Theorem 4.1.** *Suppose we run Algorithm 1 for*

$$K \geq \frac{c'_{H,c_q,c_x,c_\Theta,c_w} \cdot dp^2 \cdot \log^3(dK\delta^{-1})}{\epsilon^2}$$

*episodes, for some parameter* $c'_{H,c_q,c_x,c_\Theta,c_w}$ *depending polynomially on* $H, c_q, c_x, c_\Theta, c_w$, *Then with probability at least* $1 - \delta$, *we have for* $\widetilde{\pi}_{C,D}$ *be defined in Equation 9.*

$$\mathbb{E}_{[C,D]\sim\mu}\Big[ \mathbb{E}_{\widetilde{\pi}_{C,D}} \Big( J_1^{\widetilde{\pi}_{C,D}}([\Theta_*C, \Theta_*D], x_1) \Big) - J_1^*([\Theta_*C, \Theta_*D], x_1) \Big] \leq \epsilon. \tag{13}$$

Theorem 13 states after playing polynomial number of episodes, our agent can learn a decoder $\widetilde{\Theta}$ such that given a new LQR with contexts $(C, D)$, this decoder can turns the contexts into a near-optimal policy $\widetilde{\pi}_{C,D}$ *without any training* on the new LQR. Note this is the desired agent we want to build as described in the introduction. We emphasize again that this is the first provably efficient algorithm that builds a decoder for continuous control environments.

**Remark 4.1.** *Via similar analysis, it is easy to show that if the output* $\widetilde{\Theta}$ *is picked uniformly at random from* $\{\Theta^{(k)}\}_{k\in[K]}$, *the policy achieves similar accuracy.*

In fact, Theorem 4.1 is implies by the following regret bound of our algorithm.

**Proposition 4.1.** *With probability at least* $1 - \delta$,

$$\text{Regret}(KH) \leq c'_H \cdot d^{1/2}p \cdot \log^{3/2}(dKHc_x\delta^{-1}) \cdot \sqrt{KH}.$$

*where* $c'_H$ *is a constant depending only polynomially on* $H, c_q, c_x, c_M, c_w$.

By the definition of regret, this proposition justifies that the performance of the agent actually improves as it sees more environment.

## 5 EXPERIMENTS

In this section, we validate the effectiveness of our algorithm via numerical simulations.

We perform experiments on a path-following task. In this task, we are given a trajectory $z_1^*, z_2^*, \ldots, z_H^* \in \mathbb{R}^2$. Our goal is to exert forces $u_1, u_2, \ldots, u_m \in \mathbb{R}^2$ on objects with different (measurable) masses to minimize the total squared distance $\sum_{h=1}^H \|z_h - z_h^*\|^2 + \|u_i\|_2^2$. Each state $x_h = [z_h; v_h] \in \mathbb{R}^4$ is a vector whose first two dimensions represent the current position and the last two dimension represent the current velocity. In each stage $h$, we may exert a force $u_h \in \mathbb{R}^2$ on the object, which produces an accelerations $\frac{u_h}{m} \in \mathbb{R}^2$. The dynamics of the system can be described as

$$\begin{cases} z_{h+1} = z_h + v_h \\ v_{h+1} = k \cdot v_h + u_h/m \end{cases} \tag{14}$$

where $0 < k \leq 1$ is the decay rate of velocity induced by resistance. In our setting, the decay rate of velocity $k$ is fixed (encoded in $\Theta_*$), where the mass of the object $m$ is drawn from the uniform distribution over $[0.1, 10]$. In our experiments, we set the noise vector $w_h$ in the dynamics of the LQR system (cf. Equation 1) to be a Gaussian random vector with zero mean and covariance $10^{-4} \cdot I$. In each episode, we receive an object with mass $m$ where $m$ is draw from the uniform distribution over $[0.1, 10]$, train one trajectory using that object, and the goal is to recover the physical law described in Equation 14 so that our model can deal with objects with unseen mass $m$. Please see Appendix B for the concrete value of $\Theta_*$, $Q$ and $R$ and the distribution of $C$ and $D$.

In our experiments, we use 100 different masses as *training masses* (fixed among all experiments), and use 100 different masses as *test masses* (again fixed among all experiments). All the training masses and test masses are drawn from the uniform distribution over $[0.1, 10]$. We implement a practical version of Algorithm 1. In particular, instead of solving the optimization problem in Equation 10 exactly, we sample 100 different $\Theta$ from $\mathcal{C}^{(k)}$ uniformly at random, and choose the $\Theta$ which minimizes the objective function. Moreover, instead of using the theoretical bound for $\beta^{(k)}$ in Equation 12, we treat $\beta^{(k)}$ as a tunable parameter and set $\beta^{(k)} = 10^4$ in our experiments to encourage exploration at early stage of the algorithm. We use two different metrics to measure the accuracy of the learned model. First, we use $\|\Theta_k - \Theta_*\|_F$ where $\Theta_k$ is calculated in Line 15 to measure the accuracy of the learned $\Theta$. Moreover, using the learned $\Theta$, we test on 100 objects whose masses are the 100 test masses to calculate the control cost $\sum_{h=1}^H \|z_h - z_h^*\|^2 + \|u_i\|_2^2$. We compare the control cost of the learned $\Theta$ and the optimal control cost, and use the mean value of the differences (named mean control error) to measure the accuracy.

In all experiments we fix $H = 20$. We use three different types of trajectories: unit circle, parabola $y = x^2$ with $x \in [0, 1]$ and Lemniscate of Bernoulli with $a = 1$[1]. For all three types of trajectories we use their parametric equation $x = x(t)$ and $y = y(t)$, divide the interval $[0, 1]$ evenly into $H$ parts, and set $t$ to be the endpoints of these parts. We use these $t$ values to define the trajectory $z_1^*, z_2^*, \ldots, z_H^* \in \mathbb{R}^2$. We set the decay ratio $k$ to be $k = 1$ or $k = 0.7$ in our experiments.

We plot the accuracy of the learned model in Figure 1. Here we vary the number of training episodes (the number of training masses) and observe its effect on the accuracy. It can be observed that our algorithm achieves an satisfactory accuracy using only 5 episodes. We also illustrate trajectories obtained by our resulting controllers in Figure 2. From Figure 2, it is clear that as the agent plays more environments, it can enjoy better performance.

## 6 CONCLUSION

In this paper, we give a provably efficient algorithm for learning LQR with contexts. Our result bridges two major fields, learning with contexts and continuous control from a theoretically-principled view. For future work, it is interesting to study more complex settings, include non-linear control. Another interesting direction is to design provable algorithm in our setting with safety guarantees (Dann et al., 2018).

---

[1] https://en.wikipedia.org/wiki/Lemniscate_of_Bernoulli.

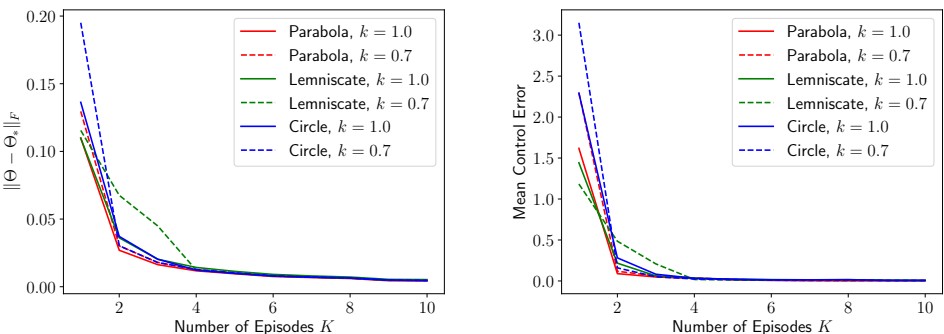

Figure 1: $\|\Theta - \Theta_*\|_F$ and Mean Control Error.

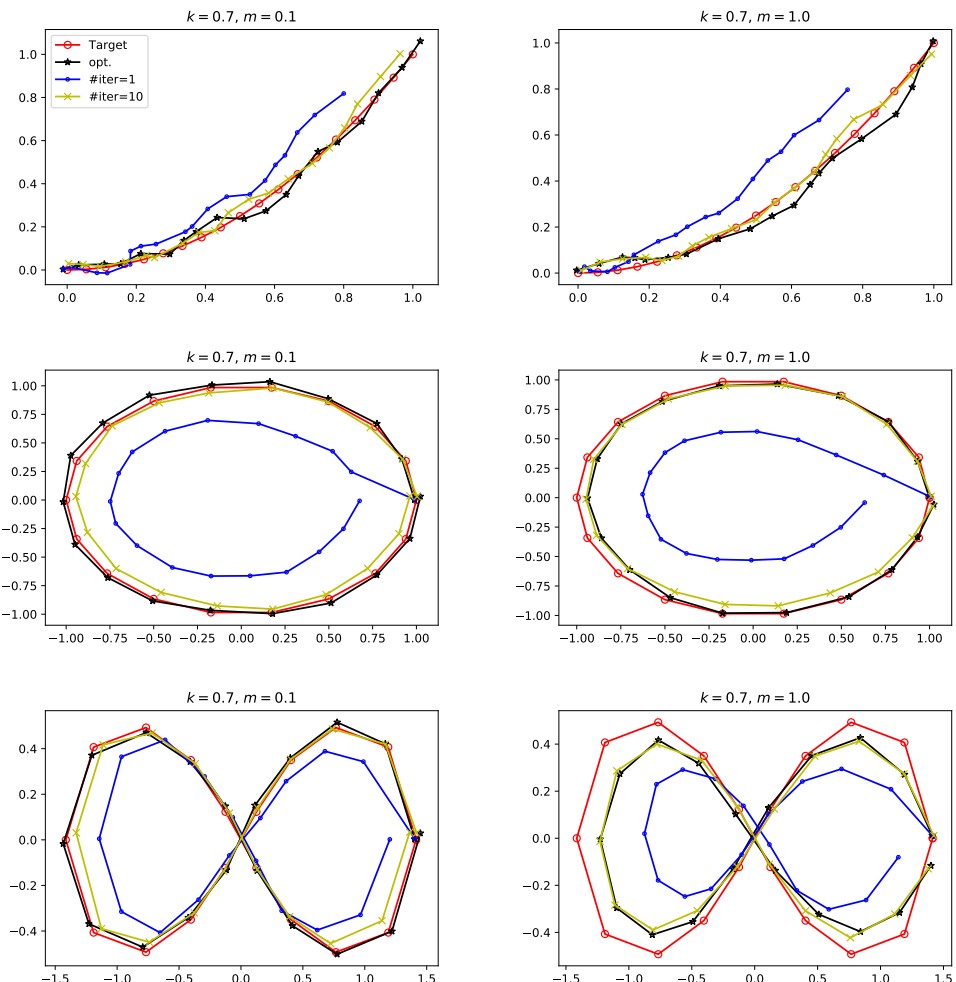

Figure 2: Example trajectories produced by the LQR controllers. We test the LQR policy to follow three types of paths: parabola, circle, and lemniscate. We first train a decoder, then test it on systems with $m = 0.1, k = 0.7$ (left column), and $m = 1.0, k = 0.7$ (right column). Dashed line with circles:target trajectories. $\star$: optimal policy. $\circ$: decoder trained by 1 iteration on randomly drawn contexts. $\triangle$: decoder trained by 3 iterations on randomly drawn contexts. $\times$: decoder trained by 10 iterations on randomly drawn contexts.

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

# A    PROOF OF MAIN RESULTS

This sections devotes to proving the main results. Before we prove Proposition 4.1, let us use it to prove Theorem 4.1.

*Proof of Theorem 4.1.* We rewrite the Equation equation 13 as follows.

$$\mathbb{E}_{C,D}\mathbb{E}_{\widetilde{\pi}}\Big[J_1^{\widetilde{\pi}}\big(M_{\Theta_*,C,D},\, x_1\big)\Big] - \mathbb{E}_{C,D}\Big[J_1^*\big(M_{\Theta_*,C,D},\, x_1\big)\Big]$$

$$= \frac{1}{K}\sum_{k=1}^{K}\mathbb{E}_{C,D}\Big[J_1^{\pi^k}\big(M_{\Theta_*,C,D},\, x_1\big)\Big] - \mathbb{E}_{C,D}\Big[J_1^*\big(M_{\Theta_*,C,D},\, x_1\big)\Big]$$

$$= \frac{1}{K}\sum_{k=1}^{K}\Big(\mathbb{E}_{C,D}\Big[J_1^{\pi^k}\big(M_{\Theta_*,C,D},\, x_1\big)\Big] - J_1^{\pi^k}\big(M_{\Theta_*,C^{(k)},D^{(k)}},\, x_1\big) + J_1^{\pi^k}\big(M_{\Theta_*,C^{(k)},D^{(k)}},\, x_1\big)$$

$$- J_1^*\big(M_{\Theta_*,C^{(k)},D^{(k)}},\, x_1\big) + J_1^*\big(M_{\Theta_*,C^{(k)},D^{(k)}},\, x_1\big) - \mathbb{E}_{C,D}\Big[J_1^*\big(M_{\Theta_*,C,D},\, x_1\big)\Big]\Big)$$

$$= R_1 + R_2 + R_3$$

where

$$R_1 = \frac{1}{K}\sum_{k=1}^{K}\Big(\mathbb{E}_{C,D}\big[J_1^{\pi^k}\big(M_{\Theta_*,C,D},\, x_1\big)\big] - J_1^{\pi^k}\big(M_{\Theta_*,C^{(k)},D^{(k)}},\, x_1\big)\Big),$$

$$R_2 = \frac{1}{K}\sum_{k=1}^{K}\Big(J_1^*\big(M_{\Theta_*,C^{(k)},D^{(k)}},\, x_1\big) - \mathbb{E}_{C,D}\big[J_1^*\big(M_{\Theta_*,C,D},\, x_1\big)\big]\Big),$$

and

$$R_3 = \frac{1}{K}\sum_{k=1}^{K}\Big(J_1^{\pi^k}\big(M_{\Theta_*,C^{(k)},D^{(k)}},\, x_1\big) - J_1^*\big(M_{\Theta_*,C^{(k)},D^{(k)}},\, x_1\big)\Big).$$

Let $\mathcal{F}_k$ be the filtration of fixing all randomness before episode $k$. We have $R_1$ and $R_2$ are Martingale difference sum. Note that the magnitude of each summand in $R_1$ or $R_2$ is upper bounded by (proved in Lemma A.3 and A.4),

$$Hc_q c_x$$

almost surely. Therefore, by Azuma's inequality (Theorem A.1), we have, with probability greater than $1 - \delta/2$,

$$|R_1| + |R_2| \le 2Hc_q c_x \cdot \sqrt{\frac{2\log(8/\delta)}{K}}.$$

Moreover, by Proposition 4.1, we have with probability greater than $1 - \delta/2$,

$$|R_3| \le c \cdot d^{1/2}p \cdot \log^{3/2}(dKHc_x^2\delta^{-1}) \cdot \sqrt{\frac{H}{K}},$$

where $c$ is constant depending only polynomially on $H$, $c_q, c_x, c_M$, and $c_w$. Combining the above two inequalities, and setting $K$ appropriately, we complete the proof of Theorem 4.1. $\qquad\square$

## A.1    USEFUL CONCENTRATION BOUNDS

Before we prove the main proposition, we first recall some useful concentration bounds.

**Theorem A.1** (Azuma's inequality)**.** *Assume that $\{X_s\}_{s\ge 0}$ is a martingale and $|X_s - X_{s-1}| \le c_s$ almost surely. Then for all $t > 0$ and all $\epsilon > 0$,*

$$\Pr\big[|X_t - X_0| \ge \epsilon\big] \le 2\exp\left(\frac{-\epsilon^2}{2\sum_{s=1}^{t}c_s^2}\right).$$

**Theorem A.2** (Martingale Concentration, Theorem 16 of Abbasi-Yadkori & Szepesvári (2011)). *Let $\mathcal{F}_t; t \geq 0$ be a filtration, $(z_t; t \geq 0)$ be an $\mathbb{R}^d$-valued stochastic process adapted to $(\mathcal{F}_t)$. Let $(\eta_t; t \geq 1)$ be a real-valued martingale difference process adapted to $\mathcal{F}_t$. Assume that $\eta_t$ is conditionally sub-Gaussian with constant L, i.e.,*

$$\forall \gamma > 0 \quad : \mathbb{E}[\gamma \eta_t | \mathcal{F}_t] \leq \exp(\gamma^2 L^2 / 2).$$

*Consider the following martingale*

$$S_t = \sum_{\tau=1}^{t} \eta_\tau z_{t-1}$$

*and the matrix-valued processes*

$$V_t = I + \sum_{\tau=0}^{t} z_{t-1} z_{t-1}^\top.$$

*Then for any $\delta \in (0, 1)$, with probability at least $1 - \delta$,*

$$\forall t \geq 0, \quad \|S_t\|_{V_t^{-1}}^2 \leq 2L^2 \log \left( \frac{\det(V_t)^{1/2}}{\delta} \right)$$

*where $\|S_t\|_{V_t^{-1}}^2 := S_t^\top V_t^{-1} S_t$.*

### A.2 PROOF OF PROPOSITION 4.1

In this section, we prove the main proposition. We first bound $\det(V^{(k)})$ for any $k$.

**Lemma A.1.** *For all $k \in [K]$,*

$$\det(V^{(k)}) \leq \left( 1 + kH c_x^2 / p \right)^p.$$

*Proof.* Since $V^{(k)}$ is PD, we have,

$$\det(V^{(k)}) \leq \left( \text{tr}(V^{(k)})/p \right)^p \leq \left( 1 + \sum_{k'=1}^{k} \sum_{h=1}^{H-1} \|z_h^{(k')}\|_2^2 / p \right)^p.$$

By Assumption 4.1, we have $\|z_h^{(k')}\|_2^2 \leq c_x^2$. This completes the proof. $\qquad \square$

Let us then define an event $E_k$ as follows.

**Definition A.1** (Good Event). *We define event $E_k$ as $\{\forall k' \leq k : \Theta_* \in \mathcal{C}^{(k')}\}$.*

We then show that the event $E_k$ happens with high probability.

**Lemma A.2.** *For all $k \in [K]$, we have $\Pr[E_k] \geq 1 - \delta$.*

*Proof.* Now we consider $\Theta_* - \Theta^{(k)}$. We immediately have

$$\Theta_*^\top - \Theta^{(k)\top} = \Theta_*^\top - (V^{(k)})^{-1} \left( \sum_{k'=1}^{k} \sum_{h=1}^{H-1} z_h^{(k')} (\Theta_* z_h^{(k')} + w_{h+1}^{(k')})^\top \right)$$

$$= \left( I - (V^{(k)})^{-1} \sum_{k'=1}^{k} \sum_{h=1}^{H-1} z_h^{(k')} z_h^{(k')\top} \right) \Theta_*^\top + (V^{(k)})^{-1} \sum_{k'=1}^{k} \sum_{h=1}^{H-1} z_h^{(k')} w_{h+1}^{(k')\top}.$$

Next, we have

$$(\Theta_* - \Theta^{(k)}) V^{(k)} (\Theta_* - \Theta^{(k)})^\top$$

$$= \Theta_* \left( I - (V^{(k)})^{-1} \sum_{k'=1}^{k} \sum_{h=1}^{H-1} z_h^{(k')} z_h^{(k')\top} \right)^\top V^{(k)} \left( I - (V^{(k)})^{-1} \sum_{k'=1}^{k} \sum_{h=1}^{H-1} z_h^{(k')} z_h^{(k')\top} \right) \Theta_*^\top$$

$$+ \Theta_* \left( I - (V^{(k)})^{-1} \sum_{k'=1}^{k} \sum_{h=1}^{H-1} z_h^{(k')} z_h^{(k')\top} \right)^\top \sum_{k'=1}^{k} \sum_{h=1}^{H-1} z_h^{(k')} w_{h+1}^{(k')\top}$$

$$+ \sum_{k'=1}^{k} \sum_{h=1}^{H-1} w_{h+1}^{(k')} z_h^{(k')\top} \left( I - (V^{(k)})^{-1} \sum_{k'=1}^{k} \sum_{h=1}^{H-1} z_h^{(k')} z_h^{(k')\top} \right) \Theta_*^\top$$

$$+ \sum_{k'=1}^{k} \sum_{h=1}^{H-1} w_{h+1}^{(k')} z_h^{(k')\top} (V^{(k)})^{-1} \sum_{k'=1}^{k} \sum_{h=1}^{H-1} z_h^{(k')} w_{h+1}^{(k')\top}$$

Note that $\sum_{k'=1}^{k} \sum_{h=1}^{H-1} z_h^{(k')} z_h^{(k')\top} = V^{(k)} - I$ and thus $(V^{(k)})^{-1} \sum_{k'=1}^{k} \sum_{h=1}^{H-1} z_h^{(k')} z_h^{(k')\top} = I - (V^{(k)})^{-1}$. Hence we have

$$\mathrm{tr}\left[ (\Theta_* - \Theta^{(k)}) V^{(k)} (\Theta_* - \Theta^{(k)})^\top \right]$$

$$= \|\Theta^*\|_{(V^{(k)})^{-1}}^2 + 2\mathrm{tr}\left( \Theta_* (V^{(k)})^{-1} \sum_{k'=1}^{k} \sum_{h=1}^{H-1} z_h^{(k')} w_{h+1}^{(k')\top} \right) + \left\| \sum_{k'=1}^{k} \sum_{h=1}^{H-1} z_h^{(k')} w_{h+1}^{(k')\top} \right\|_{(V^{(k)})^{-1}}^2$$

$$\leq \|\Theta_*\|_{(V^{(k)})^{-1}}^2 + 2\left\| \Theta_* \right\|_{(V^{(k)})^{-1}} \left\| \sum_{k'=1}^{k} \sum_{h=1}^{H-1} z_h^{(k')} w_{h+1}^{(k')\top} \right\|_{(V^{(k)})^{-1}} + \left\| \sum_{k'=1}^{k} \sum_{h=1}^{H-1} z_h^{(k')} w_{h+1}^{(k')\top} \right\|_{(V^{(k)})^{-1}}^2$$

where $\|X\|_V^2 := \mathrm{tr}(X^\top V X)$ and the last inequality uses Cauchy-Schwartz inequality. Notice that

$$\|\Theta_*\|_{(V^{(k)})^{-1}} \leq \|\Theta_*\|_F.$$

Moreover, we have

$$\left\| \sum_{k'=1}^{k} \sum_{h=1}^{H-1} z_h^{(k')} w_{h+1}^{(k')\top} \right\|_{(V^{(k)})^{-1}}^2 = \left\| (V^{(k)})^{-1/2} \sum_{k'=1}^{k} \sum_{h=1}^{H-1} z_h^{(k')} w_{h+1}^{(k')\top} \right\|_F^2$$

$$= \sum_{j \in [d]} \left\| (V^{(k)})^{-1/2} \sum_{k'=1}^{k} \sum_{h=1}^{H-1} w_{h+1,j}^{(k')} z_h^{(k')} \right\|_2^2$$

$$= \sum_{j \in [d]} \left\| \sum_{k'=1}^{k} \sum_{h=1}^{H-1} w_{h+1,j}^{(k')} z_h^{(k')} \right\|_{(V^{(k)})^{-1}}^2$$

By Theorem A.2, we have, for every $j \in [d]$, with probability at least $1 - \delta/d$, we have,

$$\left\| \sum_{k'=1}^{k} \sum_{h=1}^{H-1} w_{h+1,j}^{(k')} z_h^{(k')} \right\|_{(V^{(k)})^{-1}}^2 \leq 2c_w^2 \log(d \det(V^{(k)})^{1/2}/\delta).$$

By an union bound, we have, with probability at least $1 - \delta$,

$$\left\| \sum_{k'=1}^{k} \sum_{h=1}^{H-1} w_{h+1}^{(k')} z_h^{(k')} \right\|_{(V^{(k)})^{-1}}^2 \leq 2dc_w^2 \log(d \det(V^{(k)})^{1/2}/\delta).$$

Plugging to $\mathrm{tr}\left[ (\Theta_* - \Theta^{(k)}) V^{(k)} (\Theta_* - \Theta^{(k)})^\top \right]$, we have, with probability at least $1 - \delta$,

$$\mathrm{tr}\left[ (\Theta_* - \Theta^{(k)}) V^{(k)} (\Theta_* - \Theta^{(k)})^\top \right] \leq \left( c_\Theta + c_w \sqrt{2d \log(d \det(V^{(k)})^{1/2}/\delta)} \right)^2$$

$$\leq \left( c_\Theta + c_w \sqrt{2d \left( \log d + p \log(1 + kHc_x^2/p)/2 + \log \delta^{-1} \right)} \right)^2.$$

This completes the proof. □

We define $\mathbb{I}_{E_K}$ as the indicator for $E_K$ happens. We denote

$$M_*^{(k)} = M_{\Theta_*, C^{(k)}, D^{(k)}}, \quad M^{(k)} = M_{\Theta^{(k)}, C^{(k)}, D^{(k)}}, \quad \text{and} \quad y_h^{(k)} = [x_h^{(k)\top}, u_h^{(k)\top}]^\top.$$

On $E_k$, we have
$$\forall k \in [K]: \quad J_1^*(\widetilde{M}^{(k)}, x_1^k) \leq J_1^*(M_*^{(k)}, x_1^k).$$
We denote $\Delta^{(k)} := J_h^{\pi^k}(M_*^{(k)}, x_1) - J_h^*(M_*^{(k)}, x_1)$. We can rewrite *equation 7* as
$$\mathrm{Regret}(KH) = \sum_{k=1}^{K} \mathbb{I}_{E_k} \Delta^{(k)} + \sum_{k=1}^{K} (1 - \mathbb{I}_{E_k}) \Delta^{(k)},$$
where the second term is non-zero with probability less than $\delta$. For the first term, we have
$$\mathbb{I}_{E_k} \Delta^{(k)} \leq \mathbb{I}_{E_k} \left[ J_1^{\pi^k}(M_*^{(k)}, x_1) - J_1^*(\widetilde{M}^{(k)}, x_1)) \right] =: \mathbb{I}_{E_k} \cdot \widetilde{\Delta}_1^{(k)},$$
where
$$\widetilde{\Delta}_h^{(k)} = J_h^{\pi^k}(M_*^{(k)}, x_h) - J_h^*(\widetilde{M}^{(k)}, x_h).$$
Let us consider $\widetilde{\Delta}_h^{(k)}$. We denote filtration $\mathcal{F}_{k,h}$ as fixing the trajectory up to time $(k,h)$ and all $\{C^{(k')}, D^{(k')}\}_{k' \leq k}$.

We have
$$
\begin{aligned}
\widetilde{\Delta}_h^{(k)} =& x_h^{(k)\top} Q_h x_h^{(k)} + u_h^{(k)\top} R_h u_h^{(k)} + \mathbb{E}_{w_{h+1}^{(k)}}[J_{h+1}^{\pi^k}(M_*^{(k)}, x_{h+1}^{(k)}) \mid \mathcal{F}_{k,h}] \\
& - x_h^{(k)\top} Q_h x_h^{(k)} - u_h^{(k)\top} R_h u_h^{(k)} \\
& - \mathbb{E}_{w_{h+1}^{(k)}} \left[ (\widetilde{M}^{(k)} z_h^{(k)} + w_{h+1}^{(k)})^\top P_{h+1}(\widetilde{M}^{(k)})(\widetilde{M}^{(k)} z_h^{(k)} + w_{h+1}^{(k)}) \mid \mathcal{F}_{k,h} \right] \\
& - C_{h+1}(\widetilde{M}^{(k)}) \\
=& \mathbb{E}_{w_{h+1}^{(k)}}[J_{h+1}^{\pi^k}(M_*^{(k)}, x_{h+1}^{(k)}) \mid \mathcal{F}_{k,h}] \\
& - \mathbb{E}_{w_{h+1}^{(k)}} \left[ (\widetilde{M}^{(k)} z_h^{(k)} + w_{h+1}^{(k)})^\top P_{h+1}(\widetilde{M}^{(k)})(\widetilde{M}^{(k)} z_h^{(k)} + w_{h+1}^{(k)}) \mid \mathcal{F}_{k,h} \right] \\
& - C_{h+1}(\widetilde{M}^{(k)}) \\
=& \mathbb{E}_{w_{h+1}^{(k)}}[J_{h+1}^{\pi^k}(M_*^{(k)}, x_{h+1}^{(k)}) \mid \mathcal{F}_{k,h}] - J_{h+1}^{\pi^k}(M_*^{(k)}, x_{h+1}^{(k)}) + J_{h+1}^{\pi^k}(M_*^{(k)}, x_{h+1}^{(k)}) \\
& - \left(\widetilde{M}^{(k)} z_h^{(k)}\right)^\top P_{h+1}(\widetilde{M}^{(k)}) \left(\widetilde{M}^{(k)} z_h^{(k)}\right) - C_{h+1}(\widetilde{M}^{(k)}) \\
& - \mathbb{E}_{w_{h+1}^{(k)}} \left[ \left(w_{h+1}^{(k)}\right)^\top P_{h+1}(\widetilde{M}^{(k)}) w_{h+1}^{(k)} \mid \mathcal{F}_{k,h} \right] \\
=& \delta_h^{(k)} + J_{h+1}^{\pi^k}(M_*^{(k)}, x_{h+1}^{(k)}) - \left(\widetilde{M}^{(k)} z_h^{(k)}\right)^\top P_{h+1}(\widetilde{M}^{(k)}) \left(\widetilde{M}^{(k)} z_h^{(k)}\right) - C_{h+1}(\widetilde{M}^{(k)}) \\
& - \mathbb{E}_{w_{h+1}^{(k)}} \left[ \left(x_{h+1}^{(k)} - M_*^{(k)} z_h^{(k)}\right)^\top P_{h+1}(\widetilde{M}^{(k)})\left(x_{h+1}^{(k)} - M_*^{(k)} z_h^{(k)}\right) \mid \mathcal{F}_{k,h} \right] \\
=& \delta_h^{(k)} + J_{h+1}^{\pi^k}(M_*^{(k)}, x_{h+1}^{(k)}) - \left(\widetilde{M}^{(k)} z_h^{(k)}\right)^\top P_{h+1}(\widetilde{M}^{(k)}) \left(\widetilde{M}^{(k)} z_h^{(k)}\right) - C_{h+1}(\widetilde{M}^{(k)}) \\
& - \mathbb{E}_{w_{h+1}^{(k)}} \left[ \left(x_{h+1}^{(k)}\right)^\top P_{h+1}(\widetilde{M}^{(k)})\left(x_{h+1}^{(k)}\right) \mid \mathcal{F}_{k,h} \right] + \left(M_*^{(k)} z_h^{(k)}\right)^\top P_{h+1}(\widetilde{M}^{(k)})\left(M_*^{(k)} z_h^{(k)}\right) \\
=& \delta_h^{(k)} + \delta_h^{'(k)} + \delta_h^{''(k)} + J_{h+1}^{\pi^k}(M_*, x_{h+1}^{(k)}) - J_{h+1}^*(\widetilde{M}^{(k)}, x_{h+1}^{(k)})
\end{aligned}
$$
where
$$\delta_h^{(k)} = \mathbb{E}_{w_{h+1}^{(k)}}[J_{h+1}^{\pi^k}(M_*^{(k)}, x_{h+1}^{(k)}) \mid \mathcal{F}_{k,h}] - J_{h+1}^{\pi^k}(M_*^{(k)}, x_{h+1}^{(k)}) \tag{15}$$
$$\delta_h^{'(k)} = \left(x_{h+1}^{(k)}\right)^\top P_{h+1}(\widetilde{M}^{(k)})\left(x_{h+1}^{(k)}\right) - \mathbb{E}_{w_{h+1}^{(k)}} \left[ \left(x_{h+1}^{(k)}\right)^\top P_{h+1}(\widetilde{M}^{(k)})\left(x_{h+1}^{(k)}\right) \mid \mathcal{F}_{k,h} \right] \tag{16}$$
$$\delta_h^{''(k)} = \left(M_*^{(k)} z_h^{(k)}\right)^\top P_{h+1}(\widetilde{M}^{(k)})\left(M_*^{(k)} z_h^{(k)}\right) - \left(\widetilde{M}^{(k)} z_h^{(k)}\right)^\top P_{h+1}(\widetilde{M}^{(k)})\left(\widetilde{M}^{(k)} z_h^{(k)}\right). \tag{17}$$
By induction, we have
$$\sum_{k'=1}^{k} \widetilde{\Delta}_1^{(k)} \leq \sum_{k'=1}^{k} \sum_{h=1}^{H-1} \left( \delta_h^{(k)} + \delta_h^{'(k)} + \delta_h^{''(k)} \right).$$

Notice that $\delta_h^{(k)}$ and $\delta_h^{(k)}$ are Martingale difference adapted to $\mathcal{F}_{k,h}$. We can well bound the sum of them via Azuma's inequality.

**Lemma A.3.** *For all $h \in [H]$, $|J_h^{\pi^k}(M_*^{(k)}, x_h^{(k)})| \leq (H - h + 1) \cdot c_q \cdot c_x$.*

*Proof.* Prove by induction on $h$. The base case $J_H^{\pi^k}(M_*^{(k)}, x_H^{(k)}) = x_H^{(k)\top} Q_H x_H^{(k)} \leq c_q c_x^2$ holds straightforwardly. Consider an arbitrary $h < H$, we have

$$J_h^{\pi^k}(M_*^{(k)}, x_h^{(k)}) = x_h^{(k)\top} Q_h x_h^{(k)} + u_h^{(k)\top} R_h u_h^{(k)} + \mathbb{E}_{w_{h+1}^{(k)}}[J_{h+1}^{\pi^k}(M_*^{(k)}, x_{h+1}^{(k)}) \mid \mathcal{F}_{k,h}] \leq c_q c_x + (H - h) \cdot c_q \cdot c_x$$

as desired. $\square$

**Lemma A.4.** *For all $x \in X$, we have $\mathbb{I}_{E_K} |J_h^*(\widetilde{M}^{(k)}, x)| \leq c_q c_x$.*

*Proof.* Follows from Assumption 4.1. $\square$

We are now ready to prove Proposition 4.1.

*Proof of Proposition 4.1.* Thus by Azuma's inequality, we have, with probability at least $1 - \delta$,

$$\Big| \sum_{k'=1}^{k} \sum_{h=1}^{H-1} \delta_h^{(k)} \Big| \leq \sqrt{2kH \cdot [(H - h + 1)qc_x + c_q c_x]^2 \cdot \log \frac{2}{\delta}}.$$

And, with probability at least $1 - \delta$,

$$\Big| \sum_{k'=1}^{k} \sum_{h=1}^{H-1} \delta_h^{'(k)} \Big| \leq \sqrt{8kH \cdot c_x^2 c_q^2 \cdot \log \frac{2}{\delta}}.$$

For $\sum \delta_h^{''(k)}$, we bound it here.

$$\Big| \sum_{k'=1}^{k} \sum_{h=1}^{H-1} \delta_h^{''(k)} \Big| \leq \sum_{k'=1}^{k} \sum_{h=1}^{H-1} \Big| \delta_h^{''(k)} \Big| = \sum_{k'=1}^{k} \sum_{h=1}^{H-1} \Big| \|P_{h+1}(\widetilde{M}^{(k)})^{1/2}(\widetilde{M}^{(k)} y_h^{(k)})\|_2^2 - \|P_{h+1}(\widetilde{M}^{(k)})^{1/2}(M^* y_h^{(k)})\|_2^2 \Big|$$

$$\leq \sum_{k'=1}^{k} \sum_{h=1}^{H-1} \Big| \big(\|P_{h+1}(\widetilde{M}^{(k)})^{1/2}(\widetilde{M}^{(k)} y_h^{(k)})\|_2 - \|P_{h+1}(\widetilde{M}^{(k)})^{1/2}(M^* y_h^{(k)})\|_2\big)$$

$$\cdot \big(\|P_{h+1}(\widetilde{M}^{(k)})^{1/2}(\widetilde{M}^{(k)} y_h^{(k)})\|_2 + \|P_{h+1}(\widetilde{M}^{(k)})^{1/2}(M^* y_h^{(k)})\|_2\big) \Big|$$

$$\leq \Big[ \sum_{k'=1}^{k} \sum_{h=1}^{H-1} \big(\|P_{h+1}(\widetilde{M}^{(k)})^{1/2}(\widetilde{M}^{(k)} y_h^{(k)})\|_2 - \|P_{h+1}(\widetilde{M}^{(k)})^{1/2}(M^* y_h^{(k)})\|_2\big)^2 \Big]^{1/2}$$

$$\cdot \Big[ \sum_{k'=1}^{k} \sum_{h=1}^{H-1} \big(\|P_{h+1}(\widetilde{M}^{(k)})^{1/2}(\widetilde{M}^{(k)} y_h^{(k)})\|_2 + \|P_{h+1}(\widetilde{M}^{(k)})^{1/2}(M^* y_h^{(k)})\|_2\big)^2 \Big]^{1/2}$$

Notice that $\|P_{h+1}(\widetilde{M}^{(k)})^{1/2}(\widetilde{M}^{(k)} y_h^{(k)})\|_2 \leq c_q c_x c_\Theta$ and $\|P_{h+1}(\widetilde{M}^{(k)})^{1/2}(M^* y_h^{(k)})\|_2 \leq c_q c_x$. Hence

$$\Big[ \sum_{k'=1}^{k} \sum_{h=1}^{H-1} \Big| \big(\|P_{h+1}(\widetilde{M}^{(k)})^{1/2}(\widetilde{M}^{(k)} y_h^{(k)})\|_2 + \|P_{h+1}(\widetilde{M}^{(k)})^{1/2}(M^* y_h^{(k)})\|_2\big)^2 \Big| \Big]^{1/2}$$

$$\leq \sqrt{kH \cdot (c_q c_x (1 + c_\Theta))^2}.$$

Moreover, by triangle inequality, we have

$$\Big| \|P_{h+1}(\widetilde{M}^{(k)})^{1/2}(\widetilde{M}^{(k)} y_h^{(k)})\|_2 - \|P_{h+1}(\widetilde{M}^{(k)})^{1/2}(M^* y_h^{(k)})\|_2 \Big|$$

$$\leq \|P_{h+1}(\widetilde{M}^{(k)})^{1/2}(\widetilde{M}^{(k)} - M^*) y_h^{(k)}\|_2$$

$$\leq c_q \|(\widetilde{M}^{(k)} - M^*) y_h^{(k)}\|_2$$

$$\leq c_q \|(\widetilde{M}^{(k)} - M^*)(V^{(k)})^{1/2}(V^{(k)})^{-1/2} y_h^{(k)}\|_2$$

$$\leq c_q \|\big(\widetilde{M}^{(k)} - M^*\big)(V^{(k)})^{1/2}\|_2 \|(V^{(k)})^{-1/2} y_h^{(k)}\|_2$$

$$\leq c_q \cdot \sqrt{\beta^{(k)}} \cdot \|(V^{(k)})^{-1/2} y_h^{(k)}\|_2.$$

By Assumption 2, we also have $\|(V^{(k)})^{-1/2} y_h^{(k)}\|_2 \leq \|(y_h^{(k)}\|_2 \leq \sqrt{c_x}$. Hence,

$$\left| \|P_{h+1}(\widetilde{M}^{(k)})^{1/2}\big(\widetilde{M}^{(k)} y_h^{(k)}\big)\|_2 - \|P_{h+1}(\widetilde{M}^{(k)})^{1/2}\big(M^* y_h^{(k)}\big)\|_2 \right|$$

$$\leq c_q \sqrt{c_x} \cdot \sqrt{\beta^{(k)}} \cdot \min\big(\|(V^{(k)})^{-1/2} y_h^{(k)}\|_2, 1\big)$$

Combining the above equations, we have,

$$\left| \sum_{k'=1}^{k} \sum_{h=1}^{H-1} \delta_h^{''(k)} \right| \leq \sqrt{kH \cdot (c_q c_x (1 + c_\Theta))^2} \cdot c_q \sqrt{c_x} \cdot \sqrt{\beta^{(k)}} \cdot \sqrt{\sum_{k'=1}^{k} \sum_{h=1}^{H-1} \min\big(\|(V^{(k)})^{-1/2} y_h^{(k)}\|_2^2, 1\big)}$$

$$\leq 2 c_x^{3/2} c_q^2 c_\Theta \cdot \sqrt{\beta^{(k)}} \cdot \sqrt{\sum_{k'=1}^{k} \sum_{h=1}^{H-1} \log\Big(1 + \|(V^{(k)})^{-1/2} y_h^{(k)}\|_2^2\Big)} \cdot \sqrt{kH}.$$

Lastly, by Lemma 8 of Yang & Wang (2019), we have

$$\sum_{k'=1}^{k} \sum_{h=1}^{H-1} \log\Big(1 + \|(V^{(k)})^{-1/2} y_h^{(k)}\|_2^2\Big) \leq 2H \log \det(V^{(k)}).$$

Together with Lemma A.1, we have

$$2H \log \det(V^{(k)}) \leq 2Hp \cdot \log\big(1 + kHc_x^2/p\big).$$

Overall, we have,

$$\left| \sum_{k'=1}^{k} \sum_{h=1}^{H-1} \delta_h^{''(k)} \right| \leq 2 c_x^{3/2} c_q^2 c_\Theta \cdot \sqrt{2Hp \cdot \log\big(1 + kHc_x^2/p\big) \cdot (\beta^{(k)}) \cdot kH}.$$

Putting everything together, with probability at least $1 - 2\delta$, we have

$$\text{reg}(KH) \leq \sum_{k'=1}^{K} \sum_{h=1}^{H-1} \big(\delta_h^{(k)} + \delta_h^{'(k)} + \delta_h^{''(k)}\big)$$

$$\leq \sqrt{2KH \cdot \big[(H - h + 1)c_q c_x + c_q c_x\big]^2 \cdot \log \frac{2}{\delta}} + \sqrt{8KH \cdot c_x^2 c_q^2 \cdot \log \frac{2}{\delta}}$$

$$+ 2 c_x^{3/2} c_q^2 c_\Theta \cdot \sqrt{2Hp \cdot \log\big(1 + KHc_x^2/p\big) \cdot (\beta^{(K)}) \cdot KH}$$

$$\leq c_H \cdot d^{1/2} p \cdot \log^{3/2}(dKHc_x^2 \delta^{-1}) \cdot \sqrt{KH},$$

where $c_H$ is a constant depending on $H, c_q, c_x, c_\Theta$ and $c_w$. $\qquad \square$

## B  CONCRETE CHOICE OF THE PARAMETERS

We further augment the state so that the first coordinate is a constant with value 1. More specifically, we set the state $x_h = [1; z_h; v_h] \in \mathbb{R}^5$. We set

$$Q_h = \begin{pmatrix} \|z_h^*\|_2^2 & -z_h^* & 0 \\ -z_h^* & I & 0 \\ 0 & 0 & 0 \end{pmatrix}$$

so that for any state $x_h$, $x_h^T Q_h x_H = \|z_h - z_h^*\|_2^2$. We set $R_h = I$ with size $2 \times 2$. We set

$$\Theta_* = \begin{pmatrix} 1 & 0 & 0 & 0 & 0 & 0 & 0 \\ 0 & 1 & 0 & 1 & 0 & 0 & 0 \\ 0 & 0 & 1 & 0 & 1 & 0 & 0 \\ 0 & 0 & 0 & k & 0 & 1 & 0 \\ 0 & 0 & 0 & k & 0 & 1 \end{pmatrix},$$

$C$ to be the $5 \times 5$ identity matrix and $D$ to be $I/m$ with size $2 \times 2$ where $m$ is sampled from the uniform distribution over $[0.1, 10]$, to represent the physical law in Equation 14.

