# OpenReview forum: "Continuous Control with Contexts, Provably"
_ICLR.cc/2020/Conference — Reject_

### Official Review · AnonReviewer2 · 2019-10-23
**Official Blind Review #2**

**Rating:** 3

**Review:**


# Summary
- The paper proposes a UCB-inspired algorithm for a contextual LQR problem. The problem itself is introduced in this paper and is similar in spirit to CMDPs, with the difference that instead of learning a mapping from context to transition matrix, a mapping from context to matrices [A, B] figuring in the system dynamics of LQR is learned.
- The proposed algorithm is an online-learning algorithm shown to have sublinear regret in the number of experienced environments. A toy experiment with a 2D moving mass is presented to illustrate the theory.

# Decision
Although the problem setting is interesting and it is encouraging to have a guarantee, several important unclear points in the paper and a missing comparison to a straightforward baseline stop me from recommending it for publication in its present form. I detail my concerns below.

# Concerns
1) First, a conceptual question. I can see a straightforward algorithm that can learn the linear mapping \theta from context to [A, B] as follows.
        - In episode k, obtain trajectory (x_{1:H}, u_{1:H-1})
        - By least squares, find [A, B] from the obtained trajectory
        - Since context [C, D] is observed, find \theta : [C, D] -> [A, B] again by least squares
One can do this using data from K episodes if needed, one can sequentially update the controller for collecting data, etc.
=>  A comparison to such a basic approach should be definitely included in the paper, in my opinion.

2) The authors might argue that the algorithm suggested above has no guarantee. I would be curious to hear in this regard a comment on the practical implementation suggested in the paper. Namely, after deriving the bounds etc., the authors make further approximations and modifications in the practical algorithm. From my point of view, these modifications defeat the purpose of the bounds, because then only empirical evaluation can confirm that these approximations have not destroyed the analysis. Alternatively, one needs to incorporate the introduced approximation errors in the analysis. In more detail,
        - Eq. (9) is not solved exactly but by random sampling. In the 2D toy task, it may be OK, but in higher-dimensional spaces, a significant error can be introduced which is not accounted for.
        - More importantly, the UCB bound \beta in Eq. (11) is not used at all in the experiments.
            => To my understanding, it is the crucial point of UCB to use the UCB-bound. If it is not used, how should one judge the resulting algorithm?

3) This is a concern regarding clarity. I didn't get (i) if matrices [Q, R] are context-dependent or not and (ii) if the agent observes them or not. This is not clearly communicated in the text.
=> Clarify whether [Q, R] are context-dependent and observed.

# AFTER REBUTTAL
After authors' clarifications and improvements on the paper, I update my score to weak reject. The reason I am still against acceptance is the lack of stronger empirical evaluations. As R4 pointed out, some clarifications on the side of the algorithm are also required.


**Experience Assessment:**

I have read many papers in this area.

**Review Assessment: Checking Correctness Of Derivations And Theory:**

I assessed the sensibility of the derivations and theory.

**Review Assessment: Checking Correctness Of Experiments:**

I assessed the sensibility of the experiments.

**Review Assessment: Thoroughness In Paper Reading:**

I read the paper at least twice and used my best judgement in assessing the paper.

---

> ### Author Response · Authors · 2019-11-10
> **Response**
>
> Thanks for raising these questions. Please find our responses to your questions below.
> 1.	The algorithm you proposed will not work. The reason is simple, since your double least square algorithm requires [A,B] estimated to be accurate. Furthermore, it is clear from bandit and RL literature that without UCB or other exploration techniques, one cannot achieve $\sqrt{T}$ type of regret bound.
>
> 2.	- For the random sampling step in experiments, it is straightforward to show that in the low-dimensional setting using a small number of samples, one can obtain a near optimal solution. We leave devising a provably efficient computational efficient approach for the high-dimensional setting as a future work.
> - The crucial point to use UCB is that UCB can balance exploration and exploitation. In practice, the specific choice of hyper-parameters ($\beta$) is tuned to achieve the best performance. Note this is standard in bandit and RL literature. In our experiments we set $\beta=10^4$ to encourage exploration, as mentioned on Page 7.
>
> 3.	Throughout the paper, we assume [Q,R] are known. This is standard in LQR literature. We will clarify this assumption in the next version.

---

> > ### Comment · AnonReviewer2 · 2019-11-13
> > **Comments on authors' response**
> >
> > Thanks for the clarifications. I still have a few concerns.
> >
> > | The algorithm you proposed will not work. The reason is simple, since your double least square algorithm requires [A,B] estimated to be accurate.
> > That depends on how long each trajectory is. Since the system is linear, efficient system identification is feasible [1]. Furthermore, please don't take my suggested algorithm literally. What I mean is that it would be helpful to compare your proposed scheme against some baselines. And there are several naive baselines one could think about, so showing that your method outperforms them could only be advantageous.
> >
> > | the specific choice of hyper-parameters (beta) is tuned to achieve the best performance
> > What I was concerned about is that the dependence on k was dropped. In UCB, the bonus grows with time, e.g., as sqrt(log t). However, looking closer at Eq. (10), it seems that even for a fixed beta, the confidence set C_k will still change in size because V_k and theta_k are changing, and it is the confidence set and not beta which is crucial for balancing exploration/exploitation. Perhaps making it a bit more explicit in the text would help the readers.
> >
> > | we assume [Q,R] are known. This is standard in LQR literature.
> > It is also standard in LQR literature that matrices A and B are known... It would still be helpful if you clarify whether [Q, R] are context-dependent.
> >
> > [1] R. Mehra. Optimal input signals for parameter estimation in dynamic systems–survey and new results.
> > IEEE Transactions on Automatic Control, 19(6):753–768, 1974.

---

> > > ### Author Response · Authors · 2019-11-15
> > > **Response**
> > >
> > > For the first question, notice that there are two main components in our algorithm: UCB-type exploration (Line 6 in Algorithm 1) and linear regression to estimate $\Theta$ (Line 15 in Algorithm 1). Here we would like to explain the necessity of both components. Notice that the problem studied in this paper includes linear bandit [1] as a special case, and most previous algorithms for solving linear bandit do have UCB-type exploration. Moreover, linear regression is also a special case of the problem studied here (if one wants to estimate $\Theta$). Our algorithm is a careful combination of these two main components, with additional steps to deal with the underlying LQR problem. Thus, our algorithm is in fact very natural from this point of view, and any algorithm with provable guarantees do need the two components mentioned above (since it needs to be able to solve the two special cases).
> > >
> > > For the second question, our theoretical bound for $\beta^{(k)}$ does depend on $k$, but only depends on $k$ logarithmically. For the setting of the experiments considered in this paper ($k$ is at most 100), it is reasonable to ignore such a minor dependence, and thus we use a fixed value of $\beta^{(k)}$ in our experiments. Moreover, in our experiments we do use different values of $V^{(k)}$ for the confidence set $\mathcal{C}^{(k)}$ as described in Algorithm 1. Thus, our confidence set does change as $k$ changes. We will make this point more explicit in the description of the algorithm and in the experiment description.
> > >
> > > For the third question, thanks for the suggestion. We will make this clear in the paper.
> > >
> > > [1] Stochastic linear optimization under bandit feedback. Varsha Dani, Thomas P. Hayes, and Sham M. Kakade. COLT 2008

---

### Official Review · AnonReviewer1 · 2019-10-24
**Official Blind Review #1**

**Rating:** 6

**Review:**

In order to generalize the RL agent to unseen environment, in this work the authors studied the theoretical learning problem of building a decoder on top of linear continuous control using linear quadratic regulator (LQR). They presented a simple, UCB-based algorithm that refines the estimates of the encoder while doing LQR and balances  the exploration-exploitation trade-off. In the online setting, the proposed algorithm has a O(\sqrt{T}) regret bound, where T is the number of environments the agent played. This also implies after certain exploration, the agent is able to transfer the learned knowledge to obtain a near-optimal policy to an unseen environment. To justify their theoretical bounds the authors also present experiments that demonstrate the effectiveness of the algorithm.


The work of designing decoder on top of RL/control in order to generalize to new, unseen environments is very interesting, and is pretty novel to my knowledge. The problem formulation of LQR is standard until the part where the authors introduced the output matrices (C,D), which extends the fully-observable case of LQR (that is based on state feedback only) to partially observable. Leveraging the theoretical analysis of LQR, the authors extended the analysis to the setting of output feedback with particular structures of decoder matrices (C,D) sampled from decoder \mu. The algorithm proposed is quite standard in the output-feedback LQR literature (in control or RL). But the work is still interesting because to my knowledge I am not aware of general theoretical analysis of this setting (while most analysis is based on the full state feedback).
I haven't checked the proofs very carefully in the appendix, but from the description in the main paper it seems the analysis of contextual transfer learning performance is sound, and under certain regularity assumptions the authors did provide a high-probability regret bound for this contextual transfer learning problem. It would be great if the experiments are more involved as they are a bit too simple at this point, (where the unseen environment is the change in the physical constants). I also have some difficulties understanding all the dots in figure 2. Perhaps the authors can simplify the number of trajectories plotted there to make the presentation clearer. Another comment is about the current title, currently by looking at it I have no idea that is about contextual transfer learning and LQR. It would be great if that can be more specific.


**Experience Assessment:**

I have read many papers in this area.

**Review Assessment: Checking Correctness Of Derivations And Theory:**

I assessed the sensibility of the derivations and theory.

**Review Assessment: Checking Correctness Of Experiments:**

I assessed the sensibility of the experiments.

**Review Assessment: Thoroughness In Paper Reading:**

I made a quick assessment of this paper.

---

> ### Author Response · Authors · 2019-11-10
> **Response**
>
> Thanks for the positive review. We will improve and polish our experiment section in the final version. We will also update the title. Thanks for the suggestion.

---

### Official Review · AnonReviewer4 · 2019-11-20
**Official Blind Review #4**

**Rating:** 1

**Review:**

This paper considers the problem of changing environments for LQR. The authors model this through the use of a decoder that maps an incoming context (C,D) to the LQR matrices (A,B). They provide an algorithm for this setting based on a UCB strategy, prove sample complexity and regret bounds, and experimental results.

Overall the paper was well written but I had several concerns.

1. The results of this paper were not contrasted with other papers in this area. For example, if C,D are constant, and \Theta_* is a Block diagonal matrix with A,B on the diagonal - then the contextual case reduces to the standard LQR problem. It's unclear how the results given compare to past results in this setting, for example those of Abbasi-Yadkori/Szepesvari 2011.

2. I did not fully understand the UCB nature of the algorithm. In each round \Theta^(k) (the least squares estimator) seems to be used to compute the optimal policy (line 10 of the algorithm) instead \tilde{\Theta}^(k)  -the optimistic estimate. The optimistic estimate is only used in line 16 - a randomized procedure that is unmotivated.

3. Building on (1), it is hard to understand the results as given since there are no lower bounds given nor is there a discussion of the problem dependent parameters that arise. For example, in Theorem 1, is dp^2 suspected to be tight? Since the number of parameters in \Theta^{\ast} is d(p+p'), perhaps this is off by a factor of p?

3. I struggled to understand the setup of the experiments - as described the algorithm given was not used at all, rather \Theta^(k) was approximated and beta^k was set to be a constant. This does not seem like a fair evaluation of the method.

In summary, I would reject this submission unless the authors couch it better in past work, explain their results better, and improve the experiment setup.

Finally, a typo: I think the indexing variable in the equation on the top of page 4 is h' not h.


**Experience Assessment:**

I do not know much about this area.

**Review Assessment: Checking Correctness Of Derivations And Theory:**

I did not assess the derivations or theory.

**Review Assessment: Checking Correctness Of Experiments:**

I assessed the sensibility of the experiments.

**Review Assessment: Thoroughness In Paper Reading:**

I read the paper thoroughly.

---

### Author Response · Authors · 2019-11-15
**General Response and Revision Summary**

We thank both reviewers for the constructive comments! All major changes are marked in red. We made the following main changes in our revision.
We changed the title according to suggestion from Review #1.
We updated Figure 2 to make the presentation clearer according to the suggestion from Review #1.
We added a clarification that Q and R are known to the agent according to the suggestion from Review #2.
We added a paragraph to discuss a naïve approach and motivate our algorithm according to the suggestion from Review #2.
We added a clarification that V_k and C_k are changing at every epoch according to the suggestion from Review #2.

---

### Decision · Program_Chairs · 2019-12-19

**Decision:**

Reject

**Comment:**

This work considers the popular LQR objective but with [A,B] unknown and dynamically changing. At each time a context [C,D] is observed and it is assumed there exist a linear map Theta from [C,D] to [A,B]. The particular problem statement is novel, but is heavily influenced by other MDP settings and the also follows very closely to previous works. The algorithm seems computationally intractable (a problem shared by previous work this work builds on) and so in experiments a gross approximation is used.

Reviewers found the work very stylized and did not adequately review related work. For example, little attention is paid to switching linear systems and the recent LQR advances are relegated to a list of references with no discussion. The reviewers also questioned how the theory relates to the traditional setting of LQR regret, say, if [C,D] were identity at all times so that Theta = [A,B].

This paper received 3 reviews (a third was added late to the process) and my own opinion influenced the decision. While the problem statement is interesting, the work fails to put the paper in context with the existing work, and there are some questions of algorithm methods.